# More than a Bubble: Extracellular Vesicle microRNAs in Head and Neck Squamous Cell Carcinoma

**DOI:** 10.3390/cancers14051160

**Published:** 2022-02-24

**Authors:** Wittaya Panvongsa, D. Michiel Pegtel, Jens Voortman

**Affiliations:** 1Cancer Center Amsterdam, Department of Medical Oncology, Amsterdam UMC, Vrije Universiteit Amsterdam, 1081 HV Amsterdam, The Netherlands; w.panvongsa@amsterdamumc.nl; 2Cancer Center Amsterdam, Department of Pathology, Amsterdam UMC, Vrije Universiteit Amsterdam, 1081 HV Amsterdam, The Netherlands; d.pegtel@amsterdamumc.nl; 3Toxicology Graduate Program, Faculty of Science, Mahidol University, Bangkok 10400, Thailand

**Keywords:** extracellular vesicle, exosome, miRNA, head and neck squamous cell carcinoma

## Abstract

**Simple Summary:**

Head and neck squamous cell carcinoma (HNSCC) is an aggressive and lethal disease. Despite diagnostic and therapeutic advances, the overall survival of patients with advanced HNSCC remains poor. Recently, microRNAs in extracellular vesicles (EV-miRNAs) have been proposed as essential regulatory molecules involved in HNSCC. EV-miRNAs may serve as disease biomarkers and represent a novel therapeutic target. This review summarizes the current understanding of the role of EV-miRNAs in HNSCC as well as their potential future clinical applications.

**Abstract:**

MicroRNAs (miRNAs) are a class of small non-coding RNA molecules that play a pivotal regulatory role in a broad variety of biological processes. Dysregulation of miRNAs is associated with several human diseases, particularly cancer. Extracellular vesicles (EVs) are crucial components in intercellular communication. As part of the cargo of EVs, miRNAs are involved in EV-mediated cell-to-cell interactions, including promotion or suppression of tumor development. The knowledge on the molecular mechanisms and clinical importance of EV-miRNAs in head and neck squamous cell carcinoma (HNSCC) has rapidly grown over the past years. In the present review, the current understanding regarding the effect of EV-miRNAs on HNSCC tumorigenesis is summarized, which includes effects on tumor proliferation, angiogenesis, invasion and metastasis, the tumor microenvironment, immune modulation, and treatment resistance. EV-miRNA-based biomarkers in liquid biopsies such as blood and saliva may open up new possibilities for employing EV-miRNAs for screening and early diagnostics as well as disease monitoring. Future perspectives include the promise of EV-miRNAs as a novel therapeutic target.

## 1. Introduction

Head and neck squamous cell carcinoma (HNSCC) is the 8th most common malignancy worldwide, with 790,000 patients diagnosed and 400,000 patients dying from this disease each year [1]. Classical risk factors for the development of HNSCC are nicotine and alcohol abuse. High risk serotype human papillomavirus (HPV) infection is an additional risk factor for HNSCC of the oropharynx [2,3]. The prevalence of HPV-positive oropharyngeal squamous cell carcinoma (OPSCC) is considerably higher in the USA and Western countries compared to low- and middle-income countries [4]. HPV-positive OPSCC is a distinct disease entity compared to HPV-negative OPSCC, with a favorable response to treatment and better overall survival (OS) (3-year OS 82.4% versus 57.1%, respectively) [5]. Currently, testing for HPV status in OPSCC is the only prognostic molecular test used in the clinical management of HNSCC [6].

Despite attempts to improve the treatment outcome of HNSCC, the 5-year OS rate has remained unchanged over the last decade. Two-thirds of patients with HNSCC present with locoregional disease which signifies that the disease has metastasized to regional cervical lymph nodes [7,8]. Earlier stages of HNSCC are treated by surgery or radiotherapy alone, resulting in a 5-year OS of 70–90%. For locally advanced disease, multimodality treatment (a combination of surgery and/or (chemo)radiotherapy) is required with a 5-year survival rate lagging behind at 40–60% [9]. Whilst the recurrence rate for early-stage disease is 10–12%, around half of the patients with locally advanced disease experience disease recurrence within the first two years, either locoregionally or as distant metastases. Patients with recurrent and metastatic (R/M) HNSCC have a poor prognosis with a median OS of less than one year [10,11]. A more comprehensive understanding of the molecular mechanisms driving and characterizing HNSCC is urgently needed in order to improve disease outcome by better molecular diagnostics and more effective therapies.

MicroRNAs (miRNAs) are a class of small non-coding RNAs, on average 22 nucleotides in length, that play a pivotal role in the post-transcriptional regulation of gene expression. MiRNAs can induce mRNA degradation and suppression of protein translation primarily through complementary base pairing with the 3′ untranslated region of their target mRNA. Aberrant expression of miRNAs has been associated with numerous human diseases, including cancer [12].

Extracellular vesicles (EVs) are a heterogeneous population of phospholipid bilayer-enclosed particles that are released by most cell types and are widely distributed in the blood, saliva, and other body fluids [13]. EVs can be classified as exosomes or microvesicles according to their intracellular origin. The nano-sized exosomes are generated within endosomal compartments. Microvesicles bud directly from the plasma membrane [14]. EVs harbor a cargo consisting of proteins, mRNA, non-coding (nc) RNAs such as microRNAs and long nc-RNAs, DNA, and lipids. This cargo can be transferred to recipient cells, which constitutes an important form of physiological cell-to-cell communication [15]. Adversely, EVs are also involved in the development and progression of many diseases. Plasma of patients with cancer, including HNSCC, is known to be enriched in exosomes [16,17]. Over the past decade, it became clear that EVs participate in tumor progression by mediating the crosstalk between tumor cells and between stromal cells in the tumor microenvironment (TME) [14]. Notably, the EV-mediated transfer of miRNAs was shown to play a crucial role in advancing tumorigenesis by promoting angiogenesis, metastasis formation [18,19,20], TME reprogramming [21], immune tolerance [22,23], and therapy resistance [24]. By dissecting the tumor-secreted EV-miRNA profile, the importance of EV-miRNAs in cancer development is gradually being revealed. In this review, we summarize the current knowledge on EV-miRNAs in HNSCC. We also describe the potential applications of EV-miRNAs in novel treatment approaches and EV-miRNAs as diagnostic and prognostic biomarkers in HNSCC.

## 2. EV-miRNAs in Cancer

MicroRNAs (miRNAs) are a class of small non-coding RNA molecules that play pivotal regulatory roles in numerous biological processes. Dysregulation of miRNAs is associated with several human diseases, particularly cancer. Research on the role of EV-miRNAs in cancer has unveiled a broad array of mechanisms by which EV-miRNAs are implicated in carcinogenesis. Pathogenic EV-miRNAs can be actively exported by parent cells and imported by destination cells as part of vesicle trafficking and intercellular communication [25]. An ongoing debate is what determines the miRNA content of EVs. As the relative miRNA composition in EVs is different from their parent cells, an active sorting mechanism into these vesicles is suggested [25,26]. It is proposed that AGO2 and other RNA-binding proteins such as hnRNPA2B and Y-box protein 1 are involved in the regulation of miRNA loading into EVs [27,28,29]. However, further investigations, particularly in vivo experiments, are still required for more definitive conclusions [30]. EV-miRNAs can act as oncomiRs, i.e., miRNAs of which overexpression is associated with the development of cancer, or as tumor suppressor miRNAs which are generally underexpressed in cancer. EV-miRNAs can regulate cell proliferation, migration, epithelial–mesenchymal transition (EMT), tumor proliferation, angiogenesis, and metastasis formation among others. EV-miRNAs can also be employed to modulate the tumor microenvironment as well as the immune system [31]. With the ability of EV-microRNAs to regulate gene expression both locally and distantly, coinciding with the non-immunogenic character of EVs themselves, EVs may serve as a drug delivery platform for microRNA-based therapies. As EV-miRNA expression profiles are different between healthy subjects and cancer patients, they may also be used for novel diagnostic tests, including cancer screening, as well as for disease monitoring [32].

## 3. EV-miRNAs in HNSCC

The following section will be focusing on EV-miRNAs in the development and progression of HNSCC. An overview of published reports is provided in Table 1 and Figure 1.

### 3.1. EV-miRNAs in Tumorigenesis and Metastasis

Several studies have evaluated the role of EV-miRNAs in cancer cell proliferation and metastasis [50,51,52,53]. Still, many aspects concerning the exact mechanism of EV-miRNA transfer and its effects on recipient cells are unclear. Predominantly, the mechanism is reported that EV-miRNAs from donor cells modify the phenotype of recipient cells by epigenetic regulation of gene transcription. For example, Melo et al. have described how exosomes derived from patients with breast cancer can alter the transcriptome of normal cells by stimulating cell proliferation and tumor formation [54].

In HNSCC, miR-101-3p enriched exosomes derived from human bone marrow mesenchymal stem cells (hBMSCs) overexpressing miR-101-3p, were able to suppress oral cancer cell proliferation and tumor growth both in vitro and in vivo by targeting the Collagen Type X Alpha 1 Chain gene (COL10A1), resulting in downregulation of Collagen X expression [33]. A recent study demonstrated that oral squamous cell carcinoma (OSCC) derived-exosomes containing miR-130b-3p could promote angiogenesis in HUVEC cells through targeting the Phosphatase and Tensin Homolog (PTEN) tumor suppressor gene. The oncogenic effect of miR-130b-3p on tumor growth and blood vessel formation was confirmed in a tumor xenograft mouse model [34].

Invasion and metastasis formation are important aspects in HNSCC progression. It is a complex process involving cell invasion, secretion of extracellular matrix metalloproteinases (MMPs), epithelial–mesenchymal transition (EMT), and suppression of anoikis [8]. EV-miRNAs may contribute to metastasis formation in HNSCC in several ways.

First, transfer of EV-miRNAs from highly invasive tumor cells to less invasive tumor cells can induce a pro-metastatic phenotype in recipient cells. For example, release of exosomes containing miR-342–3p and miR-1246 by a highly metastatic human oral cancer cell line was found to induce cell motility and invasive abilities in poorly metastatic cells. It was hereby reported that miR-1246 directly suppressed expression of the tumor suppressor gene DENN/MADD Domain Containing 2D (DENND2D) [35]. Similarly, metastatic OSCC cells can release exosomes containing miR-200c-3p, which can bind the downstream targets chromodomain helicase DNA binding protein 9 (CHD9) and Werner syndrome RecQ like helicase (WRN), inducing an invasive phenotype in prior non-invasive OSCC cells [36]. Another study showed that OSCC-derived EVs containing miR-21-5p enhanced the metastatic phenotype of OSCC cell lines and transformed normal gingival fibroblasts (NGFs) into cancer-associated fibroblasts (CAF) [37].

As a second mechanism, certain conditions of cellular stress, such as hypoxia, can alter the miRNA composition of EVs. It is known that a high level of hypoxia is associated with poor prognosis and resistance to radiotherapy in HNSCC [55,56,57,58,59]. It is thought that under hypoxic conditions, tumor cells regulate the EV-miRNA content to modulate the tumor microenvironment and promote angiogenesis and metastasis [60,61]. For example, exosomes from hypoxic OSCC contained higher levels of oncomiRNA-21 compared to normoxic OSCC cells, which was dependent on activation of hypoxia-inducible factor (HIF)-1α and HIF-2α. MiR-21-rich exosomes induced OSCC cell migration, invasion, and expression of mesenchymal markers (Snail and Vimentin) and reduced the expression of epithelial marker E-cadherin both in vitro and in vivo [38].

Finally, tumor cells can interact with other cellular components of the tumor microenvironment via EV-miRNAs. Hsieh et al. have shown that Snail, the transcription factor regulating EMT, induces miR-21-enriched exosomes by direct transcriptional activation of the MIR21 gene. MiRNA-21 abundant exosomes promoted M2-like polarization of macrophages as well as suppression of M1-markers [39]. This shift in tumor-associated macrophage (TAM) phenotype was associated with angiogenesis and tumor growth [62]. In another study, cancer-associated fibroblast (CAF)-derived exosomes transferred miR-382-5p to OSCC cells which enhanced OSCC cell motility and invasiveness [40].

### 3.2. EV-miRNAs in the Tumor Microenvironment (TME)

In HNSCC tumors, the TME is a complex and diverse mix of tumor cells and stromal cells, including CAFs, endothelial cells, and immune cells [8]. Mesenchymal stromal cells (MSCs), major cell components in the TME, significantly influence the development and progression of cancer [63,64]. MSCs have been found to migrate into tumors and develop into tumor-associated MSCs and cancer-associated fibroblasts [65,66,67]. CAFs can greatly impact the progression of HNSCC as they can produce a wide range of growth factors (e.g., hepatocyte growth factor (HGF), epidermal growth factor (EGF), and vascular endothelial growth factor (VEGF)), cytokines (such as IL-6), matrix metalloproteinases (MMPs), and chemokines which can drive tumor cell growth, angiogenesis and immune suppression [68,69]. It has been demonstrated that EV-miRNAs originating from CAFs are essential regulators of HNSCC progression. Based on the research of Yao-Yin et al., it was found that OSCC cells gained a more aggressive phenotype after exposure to miR-34a-5p-devoid exosomes derived from CAFs. Additionally, the transfer of miR-34a-5p suppressed the proliferation and motility of OSCC cells by targeting the AXL gene (AXL receptor tyrosine kinase) which led to inhibition of the EMT-involved AKT/GSK-3β/β-catenin/Snail signaling cascade [41]. MiR-382-5p overexpression was detected in CAFs compared to fibroblasts from adjacent normal tissue. Although the exact mechanism remains unclear, miR-382-5p containing CAF-derived exosomes were suggested to be responsible for OSCC cell migration and invasion [40]. CAF-derived miR-196a-rich exosomes were proposed to play a key function in regulating HNSCC cell survival and proliferation. MiR-196a targets inhibitor of growth 5 (ING5) and cyclin-dependent kinase inhibitor 1B (CDKN1B), conferring cisplatin resistance to HNSCC cells [42]. Another study compared the differential miRNA profiles between exosomes from CAFs and normal fibroblasts (NFs). MiR-3188 was shown to be the most downregulated miRNA in CAF-derived exosomes. The loss of miR-3188 in CAF-derived exosomes increased proliferation and inhibited apoptosis in HNSCC cells by de-repressing B-cell lymphoma 2 (BCL2) expression both in vitro and in vivo. Exosomes rich in miR-3188 impaired tumor development in HNSCC xenografts [43]. It was shown that under hypoxic conditions, tumor cells can induce CAF-like differentiation of fibroblasts through the release of EV-miRNAs. The overexpression of miR-192 and miR-215 in hypoxic HNSCC-derived EVs was mediated by NF-κB and HIF-1α, respectively. EV-miR-192/215, when taken up by fibroblasts, resulted in downregulation of Caveolin-1 (CAV1) expression, a tumor suppressor gene that regulates the CAF-like differentiation of fibroblasts through inhibition of Transforming Growth Factor (TGF)-β/SMAD signaling. In turn, CAF-like differentiation mediates the progression of tumor cells through a positive feedback loop [44].

### 3.3. EV-miRNAs in Immune Modulation

HNSCC tumors are generally highly infiltrated by both tumor-infiltrating lymphocytes (TILs), e.g., B-cells, T-cells, and natural killer (NK) cells, and myeloid-lineage cells, e.g., macrophages, dendritic cells, neutrophils, and myeloid-derived suppressor cells (MDSCs). Prior studies have revealed evidence of immune cell dysfunction within the tumor microenvironment and in the peripheral blood of patients with advanced HNSCC [8]. The strong immunosuppressive effects of the TME allow tumors to evade immune surveillance. Consequently, many currently developed therapeutic strategies aim to restore the immune response, for instance, by treatment with immune checkpoint inhibitors [70,71]. The immunosuppressive milieu within the TME is mediated either directly by HNSCC tumor cells or indirectly via the stroma and chemokine-induced recruitment and polarization of immune cells such as MDSCs [72]. The role of EVs in immune modulation has been reported in various tumor types. However, there is limited data on EV-miRNA involvement in immune suppression in HNSCC. Momen-Heravi et al. reported that stimulation of an OSCC cell line with various doses of alcohol enhanced release of EVs containing oncogenic microRNAs such as miR-21 and miR-27. Subsequent exposure of monocytes with EVs from OSCC cells treated with alcohol resulted in activation of the NF-κB pathway and pro-tumorigenic reprogramming of monocytes [45]. The most abundant innate immune cells in the TME are tumor-associated macrophages (TAMs), which play a key role in tumor progression [73]. Macrophages derived from monocytes can be categorized into classically activated (M1) macrophages, which produce pro-inflammatory cytokines to eliminate pathogens, and alternatively activated (M2) macrophages, which secrete anti-inflammatory cytokines controlling tissue repair and immunosuppression [74]. It has been known that TAMs exhibiting a polarized M2 phenotype facilitate tumor growth and progression [75]. TAMs originate from monocytic precursors, which can differentiate and become activated in response to several stimuli released by tumor or stromal cells [76]. Snail-overexpressing HNSCC cells can, by secretion of miR-21-rich exosomes, promote M2-like polarization of tumor-associated macrophages by miR-21-mediated suppression of transcription of target genes such as programmed cell death protein 4 (PDCD4) and IL12A [39].

In adult human peripheral blood, γδ T cells represent a minor lymphocyte cell population comprising between ~0.5% and 16% of the total of CD3+ cells. They are also found in the gut- and skin-associated lymphoid systems and in organized lymphoid tissues [77]. γδ T cells have been shown to exhibit direct cytotoxicity against malignant cells and possess antigen-presenting properties, making them attractive candidates for tumor immunotherapy [78]. In contrast, pro-tumoral activities of γδ T cells have also been reported in several cancer types [79,80]. The precise mechanisms underlying the dual role γδ T cells are still obscure and further research is needed. The role of EV-miRNAs regarding γδ T cell function and expansion has been reported in OSCC. MiR-21 expression was significantly increased in exosomes from hypoxic OSCC cells compared to normoxic OSCC cells. In a normoxic environment, OSCC-derived exosomes could activate γδ T-cell expansion and cytotoxicity in a heat shock protein (HSP)70-dependent but dendritic cell-independent manner. In contrast, in a hypoxic environment, OSCC-derived, miR-21-rich exosomes enhanced the suppressive function of MDSCs through miR-21 mediated downregulation of PTEN levels and upregulation PD-L1 expression, which subsequently led to γδ T cell exhaustion [46]. A different study discovered that miR-9 was more abundant in exosomes from HPV positive HNSCC compared to HPV negative HNSCC. Exosomal miR-9 induced M1 polarization in macrophages through downregulation of peroxisome proliferator-activated receptor δ (PPARδ). The classically activated (M1) macrophages contain a higher level of inducible nitric oxide synthase (iNOS) and reactive oxygen species (ROS), which subsequently enhanced the radiosensitivity of HNSCC cells [47].

### 3.4. EV-miRNAs in Treatment Resistance

Cisplatin (CDDP)-based chemotherapy is an integral part of the treatment of advanced HNSCC. Cisplatin resistance, which can be intrinsic or acquired during treatment, is one of the most challenging issues in treating patients with HNSCC [81]. Understanding resistance mechanisms is essential in predicting treatment outcome and overcoming drug resistance with new therapeutic strategies. The mechanisms of cisplatin resistance are numerous, including a reduced cellular uptake of cisplatin, increased cellular efflux, enhanced DNA repair in response to cisplatin-induced DNA damage, and anti-apoptotic factors [82,83]. Growing evidence indicates that tumor-derived EV-miRNAs can confer a cisplatin resistant phenotype to recipient cells [84,85,86]. Exosomal miR-21 released by cisplatin-resistant OSCC cells can turn cisplatin-sensitive OSCC cells into cisplatin resistant, both in vitro and in vivo, by targeting tumor suppressor PTEN and programmed cell death 4 (PDCD4) [48]. The presence of cancer stem cells (CSCs) in HNSCC tumors contributes to the modulation of the TME, tumor progression, and resistance to treatment [87,88]. According to Chen et al., EVs produced by oral cancer stem cells (CSCs) may have a role in the development of cisplatin resistance. CSCs are a small subpopulation of cells within tumors with the ability of self-renewal, differentiation and tumor formation. The presence of CSCs was found in many cancer types and has been demonstrated as a driver of poor clinical outcome due to their contribution to chemotherapy resistance and metastasis [89]. CSC-derived EVs harboring miR-21-5p and other oncogenic signaling molecules, were shown to activate the PI3K/mTOR/STAT3 signaling pathway, leading to the CDDP resistance of differentiated OSCC cells. EVs released by CSCs were demonstrated to induce a CAF phenotype in normal gingival fibroblasts (NGFs), which subsequently induced a malignant phenotype in surrounding OSCC cells. In contrast, treatment with ovatodiolide (OV), a bioactive component of the Anisomeles indica plant, known for its anti-inflammatory properties, could reverse these effects. Treatment with OV resulted in a decrease in the oncogenic cargo of CSC-EVs, suppression of OSCC tumorigenesis, inhibition of NGF-CAF formation and normalization of the TME as well as restoration of cisplatin sensitivity [37]. These results suggest that disrupting EV-mediated communication between CSCs, tumor cells, and stroma might be utilized to overturn CDDP resistance in OSCC.

Docetaxel (DTX) is an anticancer drug with anti-tumor activity in numerous solid tumor types, including oral squamous cell carcinoma [90]. A recent study investigated the role of EV-miRNAs in docetaxel chemoresistance. Downregulation of miR-200c was associated with resistance to DTX and resulted in increased migration and invasion and decreased apoptosis in tongue squamous cell carcinoma (TSCC) cells. Overexpression of miR-200c in normal tongue epithelial cells (NTECs) resulted in the release of miR-200c abundant EVs. Exposure of DTX-resistant cells to miR-200c-rich EVs led to increased sensitivity to DTX in both in vitro and in vivo experiments by targeting of the tubulin beta 3 class III (TUBB3) and protein phosphatase 2 scaffold subunit Abeta (PPP2R1B) genes [49].

There is emerging evidence in various types of cancer that the TME can also induce chemoresistance through EV-miRNAs [91,92,93]. A study has shown that miR-196a containing exosomes are released from CAFs, enhancing proliferation, survival and conferring cisplatin resistance to HNSCC cells by targeting the CDKN1B and ING5 genes. Additionally, it was shown that the packaging of miR-196a into CAF-exosomes was mediated by RNA binding protein (RBP) heterogeneous nuclear ribonucleoprotein A1 (hnRNPA1) [42].

HPV-positive HNSCCs are more radiosensitive and display a significantly favorable clinical outcome over their HPV-negative counterparts. Although the underlying mechanisms are not fully understood, multiple factors are considered to contribute to radiosensitivity. The potential underlying mechanisms include DNA repair capacity, activation of tumor cell repopulation pathways and the oxygenation level in the tumor [94,95]. In addition, the immune cell population within the TME is thought to affect tumor radiosensitivity [96,97]. As mentioned in Section 3.3, the study by Tong et al. described miR-9-driven M1 macrophage polarization in HPV-positive HNSCC, resulting in enhanced tumor radiosensitivity [47]. The abovementioned findings indicate novel therapeutic avenues to overcome treatment resistance by intervening in the EV-miR balance in the tumor and TME.

## 4. EV-miRNAs in the Clinical Management of HNSCC

### 4.1. EV-miRNAs as a Therapeutic Target

Modulation of EV-miRNA-mediated vesicle trafficking constitutes a novel therapeutic target in cancer. So far, studies have focused on extracellular vesicles as a drug delivery system for miRNAs. The clinical success of conventional drug delivery systems such as peptides, polymers, lipid microparticles, and nanoparticles has been limited. Challenges include reaching the target tissue, crossing of the blood–brain barrier, and the effective engagement of intracellular targets. Furthermore, issues regarding toxicity and immunogenicity of non-natural delivery systems remain [98,99]. EVs have attracted tremendous attention in the context of biomolecule delivery platforms due to their ideal carrier system properties. The double-layered membrane of EVs protects its cargo from degradation and prolongs their circulation half-life [100]. Synthetic naked miRNAs have a short circulation half-life that could be increased by packaging into EVs, which offers protection from ribonucleases [101,102]. Another advantage is that EVs are able to traverse complex biological barriers such as the blood–brain barrier [102,103,104]. Importantly, when using autologous exosomes, minimal immunogenicity is exhibited, unlike viral gene transfer vectors or liposomes [102,105]. Using EV-miRNAs as a therapeutic approach has been investigated in various types of cancer [106,107,108]. The number of studies on HNSCC is, however, still limited (see Table 2 and Figure 2). A first study focused on γδ T cells as EV donors. Li et al. explored the use of γδ T cell-derived extracellular vesicles (γδ TDEs) as a drug delivery system (DDS) for miR-138 in the treatment of OSCC [109]. They found that overexpression of tumor suppressor miR-138 in γδ T cells resulted in the production of miR-138-rich TDEs. These γδTDEs could transfer miR-138 to OSCC cells, inhibiting tumor growth both in vitro and in vivo. Additionally, by targeting programmed cell death 1 (PD-1) and cytotoxic T-lymphocyte-associated antigen 4 (CTLA-4) in T-cells, the miR-138-rich γδTDEs also stimulated the proliferation and cytotoxicity of CD8+ T cells against OSCC cells. As such the miR-138-rich γδTDEs had a dual—direct and indirect—anti-tumoral effect on OSCC cells.

Mesenchymal stromal cells can also function as EV-donors. Oral potentially malignant disorders (OPMDs), such as erythroplakia, oral leukoplakia, and oral submucous fibrosis, are precursor lesions that may undergo malignant transformation to OSCC [111]. A study by Wang et al. investigated the effect of engineered MSC-EVs with a high copy number of miR-185 on OPMDs development. In a dimethylbenzanthracene (DMBA)-induced OPMD model, treatment with miR-185-MSC-EVs reduced the severity of inflammation as well as the grade and number of dysplastic lesions. Furthermore, there was a significant decrease in proliferative and angiogenesis markers and miR-185-EV treatment activated the apoptotic pathway through direct targeting of Akt, an upstream regulator of caspase-9 [110].

Collectively, EV-based drug delivery systems possess important advantages over conventional platforms. However, the utilization of EVs has obvious challenges as well, for example, how to load exogenous miRNAs into EVs. One method is to overexpress the desired miRNA in donor cells, followed by isolation of the miRNA-containing EVs. A second method is to load miRNAs into purified EVs directly. The main limitation of the first method is that the amount of RNA encapsulated into EVs may vary depending on the RNA species and/or sequence as well as the specific cellular mechanisms underlying RNA sorting into EVs [99]. On the other hand, direct loading of purified EVs with RNA molecules may impair their biological function as it can disrupt the EV membrane and structure [102,112]. Additionally, the use of exosomes generated in tumor cells may not be ideal for EV-based therapy as they might be carcinogenic [113]. Furthermore, the interaction of exogenous miRNAs and endogenous EVs needs to be investigated, as exemplified by the abovementioned study with γδTDEs. Nevertheless, given the intriguing and promising data, more research on EV-miRNAs as a therapeutic platform in HNSCC is warranted.

### 4.2. EV-miRNAs as a Disease Biomarker

Poor survival of locally advanced head and neck squamous cell carcinoma (LA-HNSCC) is partly due to challenges in early diagnosis as well as the lack of reliable biomarkers for predicting treatment outcome [6]. Currently, early diagnosis as well as staging relies on tissue biopsy and imaging studies [114,115]. As for tissue biopsy, limitations include the invasiveness of the procedure which impedes repeated sampling, as well as sampling bias due to the heterogeneity of the tumor [116]. Nowadays, liquid biopsies have become an attractive research method to identify the presence of cancer, therapy response, and cancer progression. A liquid biopsy involves the (molecular) analysis of a body fluid, most frequently blood. Advantages of liquid biopsies include the minimal invasiveness of the procedure, low cost, repeatability, and the comprehensive and real-time information on tumor cell evolution [117,118]. Liquid biopsies can be assessed for molecular biomarkers, including circulating tumor cells (CTCs), circulating tumor DNA (ctDNA), and EVs. This section will focus on EV-miRNAs as biomarkers in HNSCC diagnosis, response to therapy, prognosis, as well as its limitations and challenges. See Table 3 and Figure 2 for an overview of published studies.

A lot of research has focused on the potential of EV-miRNAs as diagnostic and prognostic markers, suggesting a clinical application for HNSCC-specific EV-miRNA signatures in body fluids. As an example, higher expression of serum exosomal miR-21 and homeobox transcript antisense RNA (HOTAIR) was associated with higher clinical stage and lymph node metastasis of patients with laryngeal squamous cell carcinoma (LSCC).

Moreover, the combination of increased exosomal miR-21 and HOTAIR could discriminate malignant (*n* = 52) from benign laryngeal disease (*n* = 49) with a sensitivity of 94.2% and specificity of 73.5% [119]. Similarly, another study proposed exosomal miR-21 as a diagnostic and prognostic biomarker of OSCC. It was reported that the exosomal miR-21 level was significantly higher in OSCC patients (*n* = 108) compared to healthy controls (*n* = 108) and that elevated exosomal miR-21 was associated with higher T stage and lymph node metastasis [38]. Another study indicated that the level of plasma exosomal miR-196a was higher in head and neck patients (HNC) (*n* = 74) compared to healthy donors (*n* = 30) and decreased after tumor resection, suggesting that exosomal miR-196a was released by tumor tissue. Higher exosomal miR-196a was associated with drug resistance and poor OS in patients with HNC. By determining the plasma exosomal miR-196a level, it was possible to separate patients in a chemoresistant and a chemosensitive group with a sensitivity of 85% and specificity of 70%. These data suggest that plasma exosomal miR-196a could serve as a prognostic factor and a predictor for chemoresistance in HNC patients [42]. In laryngeal squamous cell carcinoma (*n* = 6), RNA-seq analysis identified 34 upregulated and 41 downregulated serum exosomal miRNAs relative to healthy controls (*n* = 6). In the validation set (LSCC, *n* = 50, and 25 healthy controls), qRT-PCR revealed that miR-941-rich serum exosomes discriminated patients with LSCC patients from healthy controls with an area under the curve (AUC) of 0.797 [120]. Another study compared the miRNA content of EVs in the serum of patients with HPV positive OPSCC (*n* = 40), patients with gastroesophageal reflux disease (GERD; a benign inflammatory disease) (*n* = 20) and healthy controls (*n* = 20). This study used a customized miRNA-array to assess 112 miRNAs. An 11 EV-miRNA signature differentiated HPV-associated OPSCCs from healthy controls and patients with GERD with 90% sensitivity and 79% specificity [121]. More recently, Shimada et al. focused on identifying discriminating markers for differential diagnosis of a primary lung squamous cell carcinoma vs. a solitary lung metastasis from previously resected head and neck squamous cell carcinoma. The levels of miR-10a, miR-28, and miR-141 were significantly elevated in primary lung cancer compared to oligometastatic HNSCC, both in formalin-fixed paraffin-embedded (FFPE) tissue and in serum EVs. As the treatment approach and outcome differs between patients with primary early stage lung cancer vs. metastatic HNSCC, this study demonstrated that EV-miRNAs could be used as a diagnostic tool to guide clinical management [122]. In another study, a model based on EV-miR-491-5p expression was developed as a prediction tool to discriminate between patients with HNSCC and healthy controls, showing a sensitivity and specificity of 46.6% and 100%, respectively. Furthermore, it was found that the dynamic change of miRNA-491-5p pre- vs. post-treatment was associated with the 1-year disease recurrence rate (80% sensitivity and 69.23% specificity), as well as disease free survival and overall survival (HR of 2.82 and 5.66, respectively) [123]. This finding might help identify HNSCC patients who are at high risk of tumor recurrence.

Saliva is the most proximal body fluid in oral cancer. As a research specimen, it has many advantages, including the fact that it is easily accessible in a noninvasive manner, it contains a low background level of normal cell material (cells, DNA, RNA, and proteins), and inhibitory substances are less abundant [132]. The potential use of saliva-derived exosomal miRNAs for the detection of HNSCC has been reported. Saliva-derived exosomal miR-512-3p and miR-412-3p were upregulated, and miR-302b-3p and miR-517b-3p selectively enriched in EVs in a study of 21 OSCC patients and 11 healthy volunteers; ROC analysis showed high diagnostic power with AUC values of 0.847 and 0.871, respectively [124]. Langevin et al. sequenced exosomal miRNAs from four HNSCC cell lines, followed by a validation study using the saliva of patients with HNSCC. The results demonstrated that the levels of miR-486-5p, miR-486-3p and miR-10b-5p are increased in saliva of HNSCC patients (*n* = 11) relative to healthy controls (*n* = 9) [125]. He et al. found that the expression level of miR-24-3p in salivary exosomes from OSCC patients (*n* = 4) was substantially higher than in healthy controls (*n* = 4) (fold change 121.54). In the validation cohort, miR-24-3p could discriminate between OSCC patients (*n* = 45) and healthy controls (*n* = 10) with a sensitivity and specificity of 64.4% and 80%, respectively [126]. In another study, miR-200a and miR-134 were significantly dysregulated in OSCC patients (*n* = 14) compared to smokers (*n* = 17) and healthy controls (*n* = 6) [127].

Additionally, in vitro studies have identified several other EV-miRNA biomarkers in HNSCC. Erlotinib is a small-molecule inhibitor of the epidermal growth factor (EGFR) pathway, a known molecular target in HNSCC. However, there has been limited therapeutic success from EGFR inhibition in HNSCC with EGFR-targeting agents achieving response rates of about 4–15%. Almost all patients eventually develop resistance, suggesting innate and acquired resistance to EGFR inhibition [133,134,135,136]. Previous research examined the differentially expressed miRNA in EVs released from erlotinib-resistant and erlotinib-sensitive cells. In EVs released from erlotinib-resistant HNSCC cells, miR-7704, miR-21-5p, and miR-3960 were significantly upregulated. Transfection of these three miRNAs induced a pro-tumorigenic effect in cell lines. Inversely, let-7i-5p, miR-619-5p, and miR-30e-3p were downregulated in resistant cells. Transfection of these miRNAs induced an anti-tumor effect in cell lines. These results indicate that profiling of EV-miRNAs can potentially predict erlotinib response in HNSCC [128]. The same research group reported in another study that oncogenic miRNA-365 promotes OSCC cells’ progression [137] and is exported into exosomes, suggesting the potential role of EV-miRNA-365 as a biomarker for OSCC [129]. Another recent study identified dysregulated exosomal miRNAs comparing OSCC-derived cell lines HSC-2, HSC-3, Ca9-22, and HO-1-N1 and human normal oral keratinocytes (HNOKs) using an miRNA array. The four dysregulated miRNAs, miR-125b-5p, miR-17-5p, miR-200b-3p and miR-23a-3p, were reported as potential biomarkers for OSCC [130]. Small RNA sequencing was performed by Peacock et al. to identify an miRNA signature associated with EVs originating from HPV positive vs. HPV negative OPSCC cells. The analysis revealed that 14 miRNAs were enriched in EVs from HPV positive cells, while 19 miRNAs were enriched in EVs from HPV negative cells. These findings suggest that EV miRNAs could be used for oropharyngeal cancer subtype classification [131].

Overall, EV-miRNA profiling provides a compelling research focus with tremendous promise. However, there are still substantial challenges to overcome before profiling of EV-miRNAs will be integrated into routine clinical use. For instance, the EV isolation and purification methods used can greatly influence the results of subsequent analyses and therefore consistency in methodology is crucial. For example, ultracentrifugation (UC) is less suitable for clinical application because the procedure may disrupt EVs when high g-forces (>100,000 g) are used [138]. Using a density gradient-based protocol typically produces purer EV fractions but is time-consuming and has a lower yield [139]. A precipitation approach produces a higher particle yield, albeit less pure with a low particle-to-protein ratio [140]. Size exclusion chromatography (SEC) has been demonstrated to retain the functional properties of EVs better than UC-isolated EVs [141]. As a disadvantage, the SEC-based procedure often requires concentration steps to concentrate the dilution of EV samples resulting from ultrafiltration.

Another important challenge is the presence of significant levels of endogenous EV-miRNAs in body fluids, which hinders the identification of tumor-specific EV-miRNAs. The level of exosomal miRNAs in the circulation also fluctuates between individuals, indicating inter-individual differences. The diagnostic performance of saliva as a biomarker source appears to be tumor site-dependent, being most efficient for oral cavity cancer, as shown in ctDNA studies [114,142]. In the end, it may also be necessary to evaluate a combination of several body fluids in order to improve biomarker performance.

## 5. Conclusions and Future Perspectives

As illustrated in this review, EV-miRNAs are involved in all aspects of tumor development in HNSCC. Despite the rapidly expanding knowledge on EV-miRNAs, a lot is still unknown regarding the exact mechanisms governing this crucial form of cell-to-cell communication. Furthermore, aside from HPV status, based on transcriptomic profiling different head and neck cancer molecular subtypes may be identified with differential involvement of EV-miRNAs in disease biology.

EV-miRNAs are promising biomarkers in HNSCC, especially for (early) disease detection and prediction of treatment outcome. The limited overlap in candidate biomarker EV-miRNAs between studies is likely to reflect methodological issues in this budding field of research. Additionally, most of the studies were undertaken in small patient cohorts. Exploration and validation in large sample cohorts, in multi-center studies, using standardized protocols and analysis methods are required to prove the value of EV-miRNAs in a clinical setting. In this regard, it is necessary to include liquid biopsy analyses in clinical studies (two examples of lung cancer studies can be found on ClinicalTrials.gov: identifiers NCT04427475 and NCT03542253). To further advance the EV-miRNA field, an adequate infrastructure for collection, storage and processing of liquid biopsy specimens is needed, which can be achieved through the set-up of so-called Liquid Biopsy Centers (for example, the Cancer Center Amsterdam Liquid Biopsy Center, http://www.liquidbiopsycenter.nl/, accessed on 20 January 2022).

In conclusion, EV-miRNAs hold great promise as disease biomarkers in HNSCC. Furthermore, EV-miRNAs are a likely novel therapeutic target in HNSCC. Still in the stage of promise, but now riding on the wave of the liquid biopsy revolution, EV-miRNAs will prove to be more than a bubble in the coming future.

## Figures and Tables

**Figure 1 cancers-14-01160-f001:**
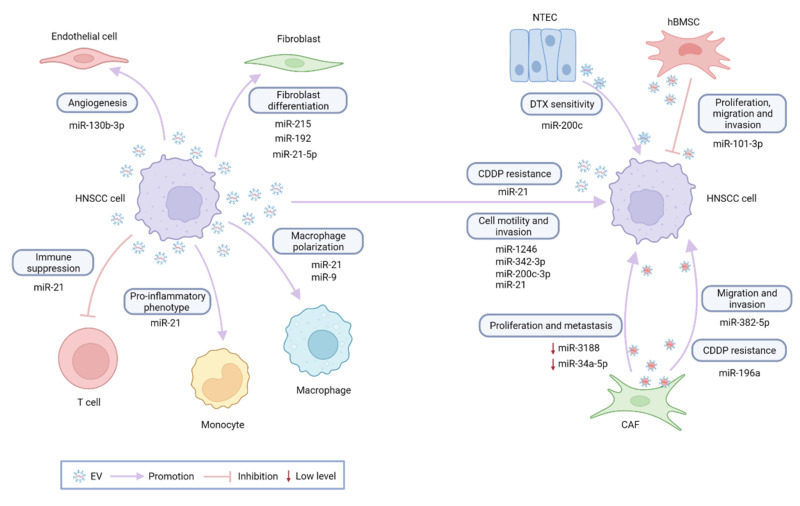
EV-miRNAs in HNSCC. As key players in an intricate tumor network, EV-miRNAs are involved in virtually all aspects of tumor development. NTEC, normal tongue epithelial cells; hBMSCs, human bone marrow mesenchymal stem cells; CAF, cancer-associated fibroblasts; CDDP, cisplatin; DTX, docetaxel.

**Figure 2 cancers-14-01160-f002:**
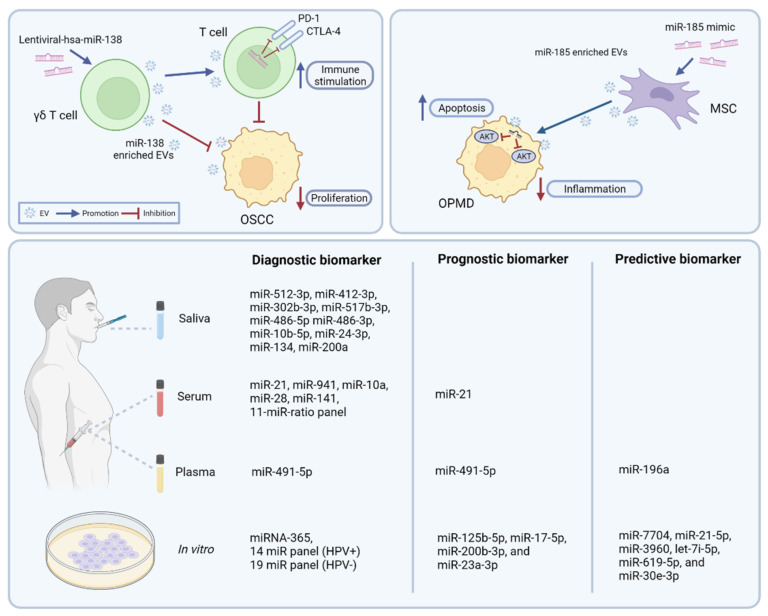
Clinical applications of EV-miRNAs in HNSCC. Upper panels display EV-miRNAs as a drug delivery system for miRNA-based therapy. The lower panel provides an overview of EV- miRNA-based candidate biomarkers for diagnosis, prognosis, and treatment response in HNSCC. CTLA-4, cytotoxic T-lymphocyte-associated antigen 4; MSC, mesenchymal stromal cell; OPMD, oral potentially malignant disorders; OSCC, oral squamous cell carcinoma; PD-1, programmed cell death 1.

**Table 1 cancers-14-01160-t001:** EV-miRNAs in HNSCC.

EV-miRNA	Donor Cell	Recipient Cell	EVs Isolation Method	miRNA Detection Method	Target	Effect	Ref.
miR-101-3p	hBMSCs	OSCC	Ultracentrifugation	Microarray and qRT-PCR	COL10A1	Inhibition of cell proliferation, invasion, and migration	[33]
miR-130b-3p	OSCC	HUVEC	Precipitation	qRT-PCR	PTEN	Promotion of tumor growth and blood vessel formation	[34]
miR-1246 miR-342–3p	Highly metastatic OSCC	Poorly metastatic OSCC	Size-exclusion chromatography	Microarray and qRT-PCR	DENND2D	Induction of cell motility and invasion	[35]
miR-200c-3p	Highly invasive OSCC	Non-invasive OSCC	Precipitation and targeted filtration	Microarray and qRT-PCR	CHD9 and WRN	Increase in invasive potential	[36]
miR-21-5p	OSCC	OSCC and NGFs	Precipitation	qRT-PCR	-	Promotion of malignant phenotype and CDDP resistance in OSCC and transformation of NGFs to CAFs	[37]
miR-21	OSCC under hypoxia	OSCC under normoxia	Precipitation	Sequencing and qRT-PCR	E-cadherin	Promotion of migration and invasion	[38]
miR-21	Hypopharyngeal SCC	Macrophages	Ultracentrifugation	qRT-PCR	PDCD4 and IL12A	Promotion of M2-like polarization of macrophages	[39]
miR-382-5p	CAFs	OSCC	Precipitation	qRT-PCR	-	Promotion of migration and invasion	[40]
miR-34a-5p	CAFs	OSCC	Ultracentrifugation and sucrose-gradient	Sequencing and qRT-PCR	AXL	Suppress OSCC cell proliferation and metastasis	[41]
miR-196a	CAFs	HNSCC	Ultracentrifugation and precipitation	Microarray and qRT-PCR	ING5 and CDKN1B	CDDP resistance	[42]
miR-3188	CAFs	HNSCC	Ultracentrifugation	Microarray and qRT-PCR	BCL2	Inhibition of cell proliferation and enhanced apoptosis	[43]
miR-215 and miR-192	OSCC under hypoxia	Fibroblast	Ultracentrifugation	Sequencing and qRT-PCR	CAV1	Induction of CAF-like phenotype	[44]
miR-21	OSCC	Monocytes	Precipitation	Microarray and qRT-PCR	-	Pro-tumorigenic reprogramming of monocytes	[45]
miR-21	OSCC under hypoxia	γδ T-cell	Precipitation	qRT-PCR	PTEN	Increased immune suppression	[46]
miR-9	HNSCC	Macrophages	Ultracentrifugation	Sequencing and qRT-PCR	PPARδ	M1-type polarization and tumor radiosensitivity	[47]
miR-21	CDDP-resistant OSCC	OSCC	Ultracentrifugation	qRT-PCR	PTEN and PDCD4	Conferment of CDDP resistance	[48]
miR-200c	TSCC	DTX-resistant TSCC	Ultracentrifugation and precipitation	Sequencing and qRT-PCR	TUBB3 and PPP2R1B	Increased sensitivity to DTX	[49]

Abbreviations: AXL, AXL receptor tyrosine kinase; BCL2, B-cell lymphoma 2; CAF, cancer-associated fibroblast; CAV1, Caveolin-1; CDDP, cisplatin; CDKN1B, cyclin-dependent kinase inhibitor 1B; CHD9, chromodomain helicase DNA binding protein 9; COL10A1, collagen type X alpha 1 chain gene; DENND2D, DENN/MADD Domain Containing 2D; DTX, docetaxel; hBMSCs, human bone marrow mesenchymal stem cells; HNSCC, head and neck squamous cell carcinoma; HUVEC, human umbilical vein endothelial cell; IL12A, interleukin 12A; ING5, inhibitor of growth 5; NGF, normal gingival fibroblast; OSCC, oral squamous cell carcinoma; PDCD4, programmed cell death protein 4; PPARδ, peroxisome proliferator-activated receptor δ; PPP2R1B, protein phosphatase 2 scaffold subunit Abeta; PTEN, phosphatase and tensin homolog; TSCC, tongue squamous cell carcinoma; TUBB3, tubulin beta 3 class III; WRN, Werner syndrome RecQ like helicase.

**Table 2 cancers-14-01160-t002:** EV-miRNA as a drug delivery system in HNSCC.

EV-miRNA	Loading Method	Donor Cell	Recipient Cell	EVs Isolation Method	miRNA Detection Method	Target	Effect	Ref.
miR-138	Lentiviral infection	γδ T cell	OSCC	Ultracentrifugation and precipitation	qRT-PCR	CTLA-4 and PD-1	Inhibition of cell proliferation and tumor growth	[109]
miR-185	miRNA-transfection	MSCs	OPMDs	Magnetic beads capturing	Microarray and qRT-PCR	AKT	Inhibition of inflammation and induction of apoptosis	[110]

**Abbreviations:** CTLA-4, cytotoxic T-lymphocyte-associated antigen 4; MSC, mesenchymal stromal cell; OPMD, oral potentially malignant disorders; OSCC, oral squamous cell carcinoma; PD-1, programmed cell death 1.

**Table 3 cancers-14-01160-t003:** EV-miRNAs as disease biomarkers for HNSCC.

EV-miRNA	Cancer Type	Source	Clinical Sample Size	EVs Isolation Method	miRNA Detection Method	Clinical Use	Ref.
miR-21	LSCC	Serum	LSCC *n* = 52 Benign laryngeal disease *n* = 49	Precipitation	qRT-PCR	Diagnosis and prognosis	[119]
miR-21	OSCC	Serum	OSCC *n* = 108 Healthy control *n* = 108	Precipitation	qRT-PCR	Prognosis	[38]
miR-196a	HNSCC	Plasma	HNSCC *n* = 74 Healthy control *n* = 30	Ultracentrifugation	Microarray and qRT-PCR	Prediction of CDDP resistance	[42]
miR-941	LSCC	Serum	LSCC *n* = 56 Healthy control *n* = 31	Ultracentrifugation and precipitation	Sequencing and qRT-PCR	Diagnosis	[120]
11-miR (ratio) panel	OPSCC	Serum	OPSCC *n* = 40, GORD *n* = 20 Healthy control *n* = 20	Precipitation	Microarray	Diagnosis	[121]
miR-10a, miR-28, and miR-141	MSQCC	serum	LSQCC *n* = 36, HNSCC *n* = 21, MSQCC *n* = 12	Magnetic beads capturing	Sequencing and qRT-PCR	Diagnosis	[122]
miR-491-5p	HNSCC	Plasma	HNSCC *n* = 73 Healthy control *n* = 20	Precipitation	Nanostring and qRT-PCR	Diagnosis and prognosis	[123]
miR-512-3p, miR-412-3p, miR-302b-3p, and miR-517b-3p	OSCC	Saliva	OSCC *n* = 21 Healthy control *n* = 11	Precipitation	qRT-PCR array and qRT-PCR	Diagnosis	[124]
miR-486-5p, miR-486-3p, and miR-10b-5p,	HNSCC	Saliva	HNSCC *n* = 11 Healthy control *n* = 9	Ultracentrifugation	Sequencing and ddPCR	Diagnosis	[125]
miR-24-3p	OSCC	Saliva	HNSCC *n* = 49 Healthy control *n* = 14	Precipitation	Microarray and qRT-PCR	Diagnosis	[126]
miR-134 and miR-200a	OSCC	Saliva	OSCC *n* = 14 Smoking group *n* = 17 Healthy control *n* = 6	Ultracentrifugation	qRT-PCR	Diagnosis	[127]
miR-7704, miR-21-5p, miR-3960, let-7i-5p, miR-619-5p, and miR-30e-3p	HNSCC	Cell lines	-	Ultracentrifugation	Microarray and qRT-PCR	Prediction of erlotinib resistance	[128]
miRNA-365	OSCC	Cell lines	-	Precipitation	qRT-PCR	Diagnosis	[129]
miR-125b-5p, miR-17-5p, miR-200b-3p, and miR-23a-3p	OSCC	Cell lines	-	Precipitation	Microarray and qRT-PCR	Prognosis	[130]
14 miRNAs (HPV+) vs. 19 miRNAs (HPV−)	OPSCC	Cell lines	-	Size exclusion chromatography	Sequencing and qRT-PCR	Diagnosis	[131]

**Abbreviations:** CDDP, cisplatin; GORD, gastroesophageal reflux disease; HNSCC, head and neck squamous cell carcinoma; LSCC, laryngeal squamous cell carcinoma; LSQCC, lung squamous cell carcinoma; MSQCC, solitary metastatic lung tumor; OPSCC, oropharyngeal squamous cell carcinoma; OSCC, oral squamous cell carcinoma.

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
