# Peer review of "More than a Bubble: Extracellular Vesicle microRNAs in Head and Neck Squamous Cell Carcinoma"

_cancers, 2022, doi:10.3390/cancers14051160_

Round 1

Reviewer 1 Report

In my view this review provides a comprehensive overview about this very interesting field of tumor biology. 

Author Response

We thank the Reviewer very much for the positive feedback regarding the manuscript.

Reviewer 2 Report

The review article by Panvongsa et al. deals with the current state of the research in the field of extracellular vesicle microRNAs (EV-miRNA) in head and neck cancer. The article is extremely well structured and deals in a very comprehensible manner with the involvement of EV-miRNAs in the areas of carcinogenesis, tumor microenvironment, immune modulation, therapy resistance and clinical application as a therapeutic and diagnostic tool. The reader gets a comprehensive insight into these still rather unknown mechanisms of miRNA interaction in head and neck cancer.

Before the manuscript is published, however, minor points should be addressed:

  • In the introduction, the general functioning of miRNAs should be briefly discussed. This would contribute to a better understanding of the following sections, which already deal with miRNA actions and interactions in detail.

  • The term oncomiRNA should be briefly explained, since not every interested reader is familiar with all aspects of miRNA (line 91).

  • Section 3.3. "EV-miRNAs in immune escpape" jumps between tumor-associated macrophages and gamma-delta T-cells, without further characterizing the importance and function of gamma-delta T-cells. This is somewhat confusing for the reader. The section should be restructured more stringently. A brief description of gamma-delta T cells follows in Section 4.1., which is very confusing.

  • Section 3.3. "EV-miRNAs in treatment resistance" reports on oral cancer stem cells (CSC). In order to better classify the importance of such cells, a brief description of the cell population should also be included here.

  • In section 4.2. "EV-miRNAs as a disease biomarker" should briefly explain the technique of liquid biopsy.

  • Furthermore, the text should be checked again for some typos (e.g. line 205, "(MDSCs))" --> "(MDSCs)")

Author Response

Response to Reviewer 2 Comments

Point 1: In the introduction, the general functioning of miRNAs should be briefly discussed. This would contribute to a better understanding of the following sections, which already deal with miRNA actions and interactions in detail.

Response 1: We thank the Reviewer for the comments on the manuscript.  As for the general functioning of miRNAs, we have now incorporated a brief section in the introduction. (Page 2, Line 57-62).

" MicroRNAs (miRNAs) are a class of small non-coding RNAs, on average 22 nucleotides in length, that play a pivotal role in the post-transcriptional regulation of gene expression. MiRNAs can induce mRNA degradation and suppression of protein translation primarily through complementary base pairing with the 3′ untranslated region of their target mRNA. Aberrant expression of miRNAs has been associated with numerous human diseases, including cancer [12].”

Point 2: The term oncomiRNA should be briefly explained, since not every interested reader is familiar with all aspects of miRNA (line 91).

Response 2: An explanation of the term oncomiRNA is now included in the revised manuscript. In addition we added an explanation of the term tumor suppressor miRNA as well (Page 3, Line 96-98).

“EV-miRNAs can act as oncomiRs, i.e. miRNAs of which overexpression is associated with the development of cancer, or as tumor suppressor miRNAs which are generally underexpressed in cancer."

Point 3: Section 3.3. "EV-miRNAs in immune escape" jumps between tumor-associated macrophages and gamma-delta T-cells, without further characterizing the importance and function of gamma-delta T-cells. This is somewhat confusing for the reader. The section should be restructured more stringently. A brief description of gamma-delta T cells follows in Section 4.1., which is very confusing.

Response 3: Thank you for helping us to improve the readability of the manuscript. Following the Reviewer’s suggestion, we have restructured section 3.3. An introduction on gamma-delta T cells has been moved from section 4.1 to section 3.3. (Page 8, Line 239-247).

“In adult human peripheral blood, γδ T cells represent a minor lymphocyte cell population comprising between 0.5% and 16% of the total of CD3+ cells. They are also found in the gut- and skin-associated lymphoid systems and in organized lymphoid tissues [73]. γδ T cells have been shown to exhibit direct cytotoxicity against malignant cells and also possess antigen-presenting properties, making them attractive candidates for tumor immunotherapy [74]. In contrast, pro-tumoral activities of γδ T cells have also been reported in several cancer types [75,76]. The precise mechanisms underlying the dual role γδ T cells are still obscure and further research is needed. The role of EV-miRNAs regarding γδ T cell function and expansion have been reported in OSCC.

Point 4: Section 3.3. "EV-miRNAs in treatment resistance" reports on oral cancer stem cells (CSC). In order to better classify the importance of such cells, a brief description of the cell population should also be included here.

Response 4: Thank you for your comment. A brief description of the CSC population was included in the revised manuscript. (Page 9, Line 275-278)

"CSCs are a small subpopulation of cells within tumors with the ability of self-renewal, differentiation and tumor formation. The presence of CSCs was found in many cancer types and has been demonstrated as a driver of poor clinical outcome due to their contribution to chemotherapy resistance and metastasis [88].”

Point 5: In section 4.2. "EV-miRNAs as a disease biomarker" should briefly explain the technique of liquid biopsy.

Response 5: Thank you for your suggestion to clarify the technique of liquid biopsy. It has now been incorporated in the manuscript (Page 13, Line 388-389).

“A liquid biopsy involves the (molecular) analysis of a body fluid, most frequently blood.”

Some comments on the different EV-isolation techniques were also added, including the limitations of the various techniques (Page 17, Line 489-499):

“For instance, the EV isolation and purification methods used can greatly influence the results of subsequent analyses and therefore consistency in methodology is crucial. For ex-ample, ultracentrifugation (UC) is less suitable for clinical application because the procedure may disrupt EVs when high g-forces (>100,000g) are used [139]. Using a density gradient-based protocol typically produces purer EV fractions but is time-consuming and has a lower yield [140]. A precipitation approach produces a higher particle yield, albeit less pure with a low particle-to-protein ratio [141]. Size exclusion chromatography (SEC) has been demonstrated to retain the functional properties of EVs better than UC isolated EVs [142]. As a disadvantage, the SEC-based procedure often requires concentration steps to concentrate the dilution of EVs samples resulting from ultrafiltration.”

Point 6: Furthermore, the text should be checked again for some typos (e.g. line 205, "(MDSCs))" --> "(MDSCs)")

Response 6: Thank you, we corrected the typo and checked the manuscript.

Reviewer 3 Report

This is a very well written article. The evidence is well presented and easy to follow.

1.I recommend adding a subheading with limitations and challenges of  extravesicular  miRNAs.

2. Interaction with host and/or tumor DNA, transcriptomic factors other than HPV status. 

Author Response

Response to Reviewer 3 Comments

Point 1: I recommend adding a subheading with limitations and challenges of extravesicular miRNAs.

Response 1: We thank the reviewer for the comments on our manuscript. We agree that it is important to highlight the limitations and challenges of EV-miRs. Instead of a separate subheading we preferred to mention limitations and challenges per section. Notably we have addressed the limitations of EV based drug delivery systems and EV-based biomarkers in sections 4.1 and 4.2 respectively. In the new version of the manuscript, we have now also included in section 4.2 a paragraph on the challenges and limitations of the various EV isolation and purification techniques (Page 17, Line 489-499).

“For instance, the EV isolation and purification methods used can greatly influence the results of subsequent analyses and therefore consistency in methodology is crucial. For ex-ample, ultracentrifugation (UC) is less suitable for clinical application because the procedure may disrupt EVs when high g-forces (>100,000g) are used [139]. Using a density gradient-based protocol typically produces purer EV fractions but is time-consuming and has a lower yield [140]. A precipitation approach produces a higher particle yield, albeit less pure with a low particle-to-protein ratio [141]. Size exclusion chromatography (SEC) has been demonstrated to retain the functional properties of EVs better than UC-isolated EVs [142]. As a disadvantage, the SEC-based procedure often requires concentration steps to concentrate the dilution of EVs samples resulting from ultrafiltration.”

  1. Interaction with host and/or tumor DNA, transcriptomic factors other than HPV status.

Response 2: We very much appreciate the comment which is referring to the complexity of EV-mediated intercellular communication and its dependence on a multitude of biological factors. We realize any review on this topic ultimately fails to be comprehensive in this regard. As the most important clinically validated molecular biomarker in head and neck cancer we did aim to review more in length the interaction of EV-miRNAs and HPV status but we realize other transcriptomic factors can equally influence EV-miRNAs. As less is known regarding these interactions we propose to include only a general remark in the current review. This would also add to the limitations comment (point 1) from the same Reviewer (Page 18, Line 511-513).

Conclusion and future perspectives:

“Furthermore, aside from HPV status, based on transcriptomic profiling different head and neck cancer molecular subtypes may be identified with differential involvement of EV-miRNAs in disease biology.”
